# Learning Infinite-Horizon Average-Reward Restless Multi-Action Bandits via Index Awareness

**Guojun Xiong, Shufan Wang, Jian Li**
SUNY-Binghamton University
{gxiong1,swang214,lij}@binghamton.edu

## Abstract

We consider the online restless bandits with average-reward and multiple actions, where the state of each arm evolves according to a Markov decision process (MDP), and the reward of pulling an arm depends on both the current state of the corresponding MDP and the action taken. Since finding the optimal control is typically intractable for restless bandits, existing learning algorithms are often computationally expensive or with a regret bound that is exponential in the number of arms and states. In this paper, we advocate *index-aware reinforcement learning* (RL) solutions to design RL algorithms operating on a much smaller dimensional subspace by exploiting the inherent structure in restless bandits. Specifically, we first propose novel index policies to address dimensionality concerns, which are provably optimal. We then leverage the indices to develop two low-complexity index-aware RL algorithms, namely, (i) `GM-R2MAB`, which has access to a generative model; and (ii) `UC-R2MAB`, which learns the model using an upper confidence style online exploitation method. We prove that both algorithms achieve a sublinear regret that is only polynomial in the number of arms and states. A key differentiator between our algorithms and existing ones stems from the fact that our RL algorithms contain a novel exploitation that leverages our proposed provably optimal index policies for decision-makings.

## 1 Introduction

Restless multi-armed bandits (`RMAB`) [64] have been used to model a variety of sequential decision making problems such as congestion control [7, 6], job scheduling [51, 35, 13, 68], wireless communication [57, 21, 9, 20, 71], healthcare [23, 14, 44, 47, 48, 40], queueing systems [28, 32, 5, 16, 42, 41], and cloud computing [17, 52, 67]. In `RMAB`, there is a collection of $N$ arms, each of which is endowed with a state that evolves independently according to a two-action Markov Decision Process (MDP) [55]. If the arm is "pulled" or "activated" at any moment in time, it advances stochastically according to one transition kernel, and if not, then it advances according to a different kernel. Rewards are generated with each transition, and the goal is to maximize the expected total reward over an infinite horizon, subject to a constraint on the number of arms activated at any moment in time.

A critical limitation of classical `RMAB` frameworks is that only two actions, either pulled or not pulled, are allowed for each arm. This is restrictive since the decision maker in many applications often has access to multiple actions for each arm [40, 18]. To this end, we consider an under-examined generalization of `RMAB` that allows for multiple actions per arm with different degrees of costs, which we call *the restless multi-action multi-armed bandits*, dubbed as `R2MAB`, in the infinite-horizon average-reward setting. Our objective is to develop simple reinforcement learning (RL) algorithms to solve `R2MAB` without knowing the underlying MDPs associated with each arm. Although `RMAB` or `R2MAB` has found its success in many applications as aforementioned, the fundamental theoretical understanding of how to develop *low computation-complexity* and *order-of-optimal regret* RL algorithms, two of

36th Conference on Neural Information Processing Systems (NeurIPS 2022).

the most important performance metrics for online restless bandits, remains in its infancy so far. In particular, three important aspects of RL algorithms for `RMAB` or `R2MAB` deserve special attentions:

▷ First, most upper confidence bound (UCB) based policies [45, 60, 46, 59] for `RMAB` often leverage *a heuristic policy* in the exploitation phase, e.g., constantly pulling one arm, which does not have any performance guarantee. Hence, these policies *may not perform close to the offline optimum*. This is partially due to the fact that finding an optimal solution for `RMAB` is PSPACE-hard [54], hence infeasible. The fundamental challenge lies in the explosion of state space and the curse of dimensionality prevents computing optimal policies. Though Whittle index policy [64] is a powerful tool to address the state space explosion, finding the Whittle index is typically intractable [51].

▷ Second, existing RL algorithms with a theoretical guarantee of a sub-linear regret upper bound, e.g., colored-UCRL2 [53], suffers from *an exponential computational complexity*, and the regret bound is *exponential in the number of states and arms*. This is due to the fact that it needs to solve Bellman equations with an exponentially large space set. Likewise, the class of Thompson sampling based algorithms [38, 37] provide theoretical guarantees in the Bayesian setting, where the updates can be computationally expensive, especially when the likelihood functions are complex.

▷ Third, although the design of low computation-complexity solutions for online `RMAB` has been gaining attentions, e.g., [25, 8, 15, 39, 62, 65, 56, 66], many challenges remain unsolved. For instance, [25, 8, 15, 39, 56] lacked finite-time performance analysis and these multi-timescale stochastic approximation algorithms often suffer from slow convergence. [65, 66] focused on the finite-horizon setting, which makes their approach not directly applicable to ours since the average reward setting studied in this paper is more challenging to analyze compared to the finite-horizon setting, which necessitates different proof techniques. [62] achieved a low-complexity policy but is constrained to a specific birth-death Markovian model and is not easy to be directly generalized.

The lack of a fundamental understanding on how to design efficient RL algorithms for `R2MAB` in the infinite-horizon average-reward setting that consider the above three aspects in terms of provably optimal policy, computational complexity, exponential pre-factor in the regret bound, and sub-linear regret performance guarantees, motivates us to fill the gap by advocating **index-aware RL solutions** in this paper. Specifically, our index-aware RL solutions operate on a much smaller dimensional subspace by exploiting the inherent structure encoded in `R2MAB` problems. This requires us to first design low-complexity provably optimal index policies for `R2MAB`, and then RL algorithms that leverage the structure of provably optimal index policies in the exploitation phase for decision-makings so as to reduce the high computational complexity and exponential factor in regret analysis. Our main contributions in this paper are summarized as follows:

• To address the dimensionality concerns in `R2MAB`, we first develop low-complexity index policies for `R2MAB` when the underlying MDPs are known. In contrast to the celebrated Whittle index policy, which requires the notoriously difficult-to-verify indexability condition, we bypass the task of deriving index policies by taking a more general linear programming (LP) approach, and hence is computationally efficient. We show that our proposed index policy is asymptotically optimal.

• We develop two index-aware RL algorithms for infinite-horizon average-reward `R2MAB`, namely, (i) `GM-R2MAB`, a generative model based approach that obtains samples initially then creates the model; and (ii) `UC-R2MAB`, an online approach wherein the model is updated as samples are obtained. Both algorithms follow a two-stage pattern of model construction and an index policy based solution. The algorithms solve an extended linear programming (ELP) problem from which we construct the provably optimal index policies to be executed during the exploitation phase in both algorithms. The key differentiator between our index-aware RL solutions and aforementioned state of the arts stems from two perspectives: (a) our index-aware RL solutions are computationally appealing since our index based solutions are merely based on solving an ELP, which is exponentially better than state-of-the-art methods such as colored-UCRL2; (b) our index-aware RL solutions contain a novel exploitation phase by leveraging our proposed provably optimal index policy for decision-makings, rather than using a heuristic one or black-box oracle in aforementioned existing algorithms.

• We provide the first-ever regret analysis for infinite-horizon average-reward `R2MAB`. We show that the above key differentiators in the design of `GM-R2MAB` and `UC-R2MAB`, contribute to their $\tilde{\mathcal{O}}(\sqrt{T})$ regret, which is only polynomial in the number of arms and states. It is worth noting that `GM-R2MAB` and `UC-R2MAB` achieve low computational complexity and a sub-linear $\tilde{\mathcal{O}}(\sqrt{T})$ regret with a polynomial pre-factor *all at once*, while none of aforementioned state of the arts achieve so.

## 2 Model Description and Problem Formulation

We consider the restless multi-action multi-armed bandits (R2MAB) problem in continuous time. There are a total of $N$ arms. Each arm $n \in \mathcal{N}$ is described by a MDP $(\mathcal{S}, \mathcal{A}, P_n, r_n)$, where $\mathcal{S}$ is the state space, $\mathcal{A}$ is the action space, $P_n(s'|s, a)$ indicates the probability of reaching state $s'$ by taking action $a$ in state $s$, and $r_n(s, a)$ is the reward of each state-action pair $(s, a)$. We assume that $\mathcal{S}$ and $\mathcal{A}$ are finite sets with cardinalities $S$ and $A$, respectively. Our results and analysis will still apply when each arm has its own sets of states and actions; so, without loss of generality, we will simply assume that all arms share the same state and action sets. We consider the unichain MDPs in this paper, which in fact is known to be necessary to guarantee the existence of stationary policies for the infinite-horizon average-reward R2MAB (and MDPs) regardless of initial states [55, 3]. We refer to action $a = 0$ as *passive action* and any action $a > 0$ as an *active action*. Moreover, using the standard terminology from the restless bandits literature [64], we call an arm *active* when an active action is applied to it and *passive* otherwise. A cost is incurred when action $a$ is applied to arm $n$ in state $s$. For abuse of notation, we denote the cost as $a$ itself. We assume that the maximum cost to activating arms at any moment in time is $B$, which we call the *activation budget*.

The decision-making scenario is as follows. At any moment in time $t$, each arm can be either active or passive. When action $a_n(t) = a$ is applied to arm $n$ in state $s_n(t) = s$, it takes an exponentially distributed amount of time to transition to state $s'$ with rate $P_n(s, a, s'), \forall s, s' \in \mathcal{S}, a \in \mathcal{A}$. Decision epochs/time are defined as the moments when the state of an arm changes. A *policy* determines what actions to be applied to each arm at each decision epoch, with the restriction that at most $B$ activation budgets can be used for activating arms. After receiving an action, each arm incurs an immediate reward $r_n(t) = r_n(s, a)$, which is a random variable with support $[0, 1]$ and mean $\bar{r}_n(s, a)$. Without loss of generality, we further assume that only active arms yield rewards, i.e., $r_n(s, 0) = 0, \forall s \in \mathcal{S}$.

**Remark 1.** *In general, there are two reward models considered in* R MAB *literature: (1) Model 1: All arms yield rewards no matter activated or not; and (2) Model 2: Only activated arms yield rewards. Both models have been widely used and risen in different applications. For example, Model 1 is widely adopted for queueing problems [16, 17, 28, 32], where all queues incur hold costs, along with others [5, 15, 25, 39]. Model 2 is widely adopted for cognitive radios [9, 20, 21, 45], where rewards are generated only on the state of selected channels, along with many other learning augmented* RMAB *settings [2, 59, 60, 62, 65]. These two models are similar without fundamental differences as discussed in [2], and they are exactly the same under the assumption that $r_n(s, 0) = 0, \forall s \in \mathcal{S}$. Our proposed solutions in this paper hold for both models.*

Let $\Pi$ be the set of all possible policies for the considered R2MAB problem and $\pi$ is a feasible policy in $\Pi$, satisfying $\pi \in \Pi : \mathcal{F}_t \mapsto \mathcal{A}^N$, where $\mathcal{F}_t$ is the sigma-algebra [58] generated by random variables $\{s_n(h), a_n(h) : \forall n \in \mathcal{N}, h \leq t\}$. The objective of the decision maker is to maximize the expected long-term average reward of activating arms subject to the activation budget constraint, i.e.,

$$\text{R2MAB:} \quad \max_{\pi \in \Pi} \quad \liminf_{T \to \infty} \frac{1}{T} \mathbb{E}_{\pi} \left( \int_{t=0}^{T} \sum_{n=1}^{N} r_n(t) dt \right), \quad \text{subject to} \sum_{n=1}^{N} a_n(t) \leq B, \quad \forall t. \quad (1)$$

When the underlying MDP (i.e., transition kernel and reward function) of each arm is known, the R2MAB problem (1) in theory can be solved using relative value iteration [55, 12]. Unfortunately, this approach suffers from the curse of dimensionality [10, 12], and lacks of insights for the solution structure. When the underlying MDPs are not known, off-the-shelf RL algorithms are either computationally expensive or without an $\tilde{\mathcal{O}}(\sqrt{T})$ regret guarantee. In the remainder of the paper, we first build a low-complexity index based policy for (1) with optimality guarantee when the MDPs are known in Section 3, and then design index-aware RL algorithms for (1) that not only are computationally efficient but also achieve an $\tilde{\mathcal{O}}(\sqrt{T})$ regret when the MDPs are unknown in Section 4.

## 3 An Index-based Policy and Asymptotic Optimality

Rather than solving (1) exactly, we instead construct an index based policy for the original R2MAB (1) that we prove to be asymptotically optimal. To describe our index policy design in Section 3.1, we first need to introduce the following linear programming (LP), in which the decision variables are the

occupancy measures [3] of the controlled MDP processes:

$$\texttt{LP}(P_n, r_n, \forall n): \quad \max_{\Omega_\pi} \sum_{n=1}^{N} \sum_{(s,a) \in \mathcal{S} \times \mathcal{A}} \omega_n(s,a) \bar{r}_n(s,a) \tag{2}$$

$$\text{subject to} \sum_{n=1}^{N} \sum_{(s,a) \in \mathcal{S} \times \mathcal{A}} a \omega_n(s,a) \leq B, \tag{3}$$

$$\sum_a \omega_n(s,a) = \sum_{s'} \sum_{a'} \omega_n(s',a') P_n(s',a',s), \quad \forall n \in \mathcal{N}, \tag{4}$$

$$\sum_{(s,a) \in \mathcal{S} \times \mathcal{A}} \omega_n(s,a) = 1, \ \omega_n(s,a) \geq 0, \quad \forall n \in \mathcal{N}, s \in \mathcal{S}, a \in \mathcal{A}. \tag{5}$$

One can arrive at the LP (2)-(5) by replacing all random variables in the relaxed version of (1) in which the activation cost at time $t$ is limited by $B$ on average, with their *expected values* via introducing a new set of variables $\omega_n(s,a)$, which are called *occupancy measures* [3] of the controlled MDP corresponding to arm $n$. Specifically, the occupancy measure $\Omega_\pi$ of a stationary policy $\pi$ for the $N$ controlled infinite-horizon MDPs is defined as the expected average number of visits to a state-action pair $(s,a)$, i.e.,

$$\Omega_\pi = \left\{ \omega_n(s,a) \triangleq \lim_{T \to \infty} \frac{1}{T} \mathbb{E}_\pi \left( \int_{t=1}^{T} \mathbb{1}(s_n(t) = s, a_n(t) = a) dt \right) : n \in \mathcal{N}, s \in \mathcal{S}, a \in \mathcal{A} \right\}.$$

It can be easily checked that the occupancy measure satisfies $\sum_{(s,a)} \omega_n(s,a) = 1$, and hence $\omega_n, \forall n \in \mathcal{N}$ is a probability measure. To this end, (4) represents the fluid transition of the occupancy measure, which holds due to the ergodic theorem [19] for finite MDPs [61]; and (5) follows the fact that the occupancy measure is a probability measure. Thus the feasible set of LP (2)-(5) is non-empty. We denote the optimal solution to the LP (2)-(5) as $\Omega_{\pi^*} = \{\omega_n^*(s,a) : n \in \mathcal{N}, s \in \mathcal{S}, a \in \mathcal{A}\}$, and the corresponding optimal value as $V^* := \sum_{n=1}^{N} \sum_{(s,a)} \omega_n^*(s,a) r_n(s,a)$, which serves as an upper bound on the reward of the original `R2MAB` (1).

**Lemma 1.** *The optimal value achieved by the LP (2)-(5) is an upper bound of that of the `R2MAB` (1).*

### 3.1 An Index Based Policy

Unfortunately, the solution to the LP (2)-(5) does not always provide a feasible decision to the original `R2MAB` (1). This is because the activation budget constraint in (1) must be met at all time, instead of just in the average sense as in (3). Exacerbating this problem is the fact that the average constraint may be violated severely during the decision epochs, resulting in poor policy performance. We overcome these challenges by introducing a computationally appealing index policy that we prove to be asymptotically optimal. Specifically, our index policy is derived from the optimal solutions $\Omega_{\pi^*} = \{\omega_n^*(s,a)\}$ and the index assigned to arm $n$ in state $s_n(t) = s$ at time $t$ is defined as

$$\mathcal{I}_n(s) := \sum_{a \in \mathcal{A}} \frac{\omega_n^*(s,a) \bar{r}_n(s,a)}{\sum_{a' \in \mathcal{A}} \omega_n^*(s,a')}, \tag{6}$$

where $\xi_n(s,a) \triangleq \frac{\omega_n^*(s,a)}{\sum_{a' \in \mathcal{A}} \omega_n^*(s,a')}$ represents the probability of applying action $a_n(t) = a$ to arm $n$ in state $s_n(t) = s$ at time $t$ [3] when $\sum_{a' \in \mathcal{A}} \omega_n^*(s,a') > 0$, and otherwise arm $n$ can be simply made passive. Hence, the index of arm $n$ in (6) represents the expected obtained reward of activating arm $n$ in state $s$. To this end, we rank all arms based on their indices (6) in a non-increasing order, and activate the set of highest indexed arms, denoted as $\mathcal{N}(t) \subset \mathcal{N}$, such that the corresponding activation cost of arms in $\mathcal{N}(t)$ is within the activation budget $B$, i.e., $\sum_{n \in \mathcal{N}(t)} a_n^\star(s_n(t)) \leq B$. Here $a_n^\star(s_n(t))$ is the action for arm $n$, which is determined according to the probability $\xi_n(s,a)$ based on its current state $s_n(t) = s$ at time $t$. When multiple arms share the same indices, we randomly activate one arm and allocate the remaining activation costs across all possible actions according to the probability $\xi_n(s,a)$. All remaining arms are kept passive in case they have zero indices. We call our index based policy, `ERC`, since it *ranks arms by **E**xpected **R**eward and pull arms constrained by activation **C**ost*. We denote the resultant index based policy as $\pi^\star = \{\pi_n^\star, n \in \mathcal{N}\}$.

**Remark 2.** *Unlike Whittle-based index policies [64, 31, 28, 72, 39, 67], our* ERC *does not require the indexability condition, which is often hard to establish especially when the transition kernel of the underlying MDP is convoluted [51]. Like Whittle policies, our* ERC *is computationally efficient since it is merely based on solving a LP. We remark that [61] also considered average-reward* R2MAB*, however, only one action can be chosen deterministically for each state with no difference in activation cost. Hence it cannot be generalized to ours since we consider a randomized policy for each state with heterogeneous activation costs across different actions. Finally, another line of works [34, 65, 66, 69, 70] designed index policies without indexability requirement for finite-horizon restless bandits, and hence cannot be directly applied to our infinite-horizon average-reward* R2MAB*.*

### 3.2 Asymptotic Optimality

We now show that ERC is asymptotically optimal in the same asymptotic regime as that in Whittle [64] and many others [63, 61, 72], where all $N$ arms are generalized to $N$ classes, and both the number of class-$n$ arms and the activation budget $B$ are scaled by $\rho$, with their ratio holding constant. Denote $X_n^\rho(\pi^\star, s, a; t)$ as the number of class-$n$ arms at state $s$ taking action $a$ at time $t$ under ERC $\pi^\star$. We will be interested in this fluid-scaling process with parameter $\rho$, and define the expected long-term average reward as $V_{\pi^\star}^\rho := \liminf_{T \to \infty} \frac{1}{T} \mathbb{E}_{\pi^\star}(\int_{t=1}^{T} \sum_{n=1}^{N} \sum_{(s,a)} r_n(s,a) \frac{X_n^\rho(\pi^\star, s, a; t)}{\rho} dt)$. Our ERC $\pi^\star$ is asymptotically optimal only when $V_{\pi^\star}^\rho \geq V_\pi^\rho, \forall \pi \in \Pi$. Before presenting our main result in this section, we first state the following technical condition called "global attractor" [63].

**Definition 1.** *An equilibrium point $X^{\rho,\star}/\rho$ under* ERC *$\pi^\star$ is a global attractor for the process $X^\rho(\pi^\star; t)/\rho$, if, for any initial point $X^\rho(\pi^\star; 0)/\rho$, the process $X^\rho(\pi^\star; t)/\rho$ converges to $X^{\rho,\star}/\rho$.*

**Remark 3.** *The global attractor indicates that all trajectories converge to $X^{\rho,\star}$. Though it may be difficult to establish analytically that a fixed point is a global attractor for the process [61], such assumption has been made in [63, 31, 61, 72, 24] and is only verified numerically. Our experimental results in Appendix E show that such convergence indeed occurs for our* ERC *$\pi^\star$.*

**Theorem 1.** *Our* ERC *$\pi^\star$ is asymptotically optimal under Definition 1, i.e., $\lim_{\rho \to \infty} V_{\pi^\star}^\rho - V_{\pi^{opt}}^\rho = 0$, where $\pi^{opt}$ represents the optimal policy for the original* R2MAB *(1).*

## 4 Index-aware Reinforcement Learning Solutions

We now consider to learn the R2MAB (1) when the transition kernel $P_n$ and reward function $r_n$, $\forall n \in \mathcal{N}$ are unknown. Our goal is to provide low-complexity index-aware RL algorithms and determine their finite-time performance measured by the regret [43], which is defined as follows:

**Definition 2.** *The regret of policy $\pi$ is defined as the expected gap between the offline optimum, i.e., the best policy under which both the transition kernels and reward functions are known, and the cumulative reward of the arm selecting algorithm, i.e., $Reg(\pi, T) := T\mu^{opt} - \mathbb{E}_\pi[R(\pi, T)]$, where $\mu(\pi) := \lim_{T \to \infty} \frac{1}{T}\mathbb{E}[R(\pi, T)] = \lim_{T \to \infty} \frac{1}{T}\mathbb{E}[\sum_{t=1}^{T} \sum_{n=1}^{N} r_n(t)]$ is the expected average reward under policy $\pi$, and $\mu^{opt} := \sup_\pi \mu(\pi)$ is the optimal average reward, which is independent of the initial state for MDPs with finite diameter [55].*

**Remark 4.** *Since finding the offline optimum for* R2MAB *is intractable, we characterize the regret with respect to the* ERC *index policy. This is due to the fact that our* ERC *index policy is asymptotically optimal. Similar definition appears in [2, 65] when learning Whittle index policy for* RMAB*.*

### 4.1 GM-R2MAB: Generative Model Based Index-Aware Reinforcement Learning

We first introduce a generative model based R2MAB learning algorithm called GM-R2MAB, which contains two phases: (i) the exploration phase (lines 1-3 in Algorithm 1) and the exploitation phase (lines 4-6 in Algorithm 1). During the exploration phase, GM-R2MAB samples each state-action pair $(s, a)$ for $J(T)$ times ($J(T)$ to be specified later) for each arm $n$, counts the number of times $T_n(s, a, s')$ for each transition to the next state $s'$, and constructs an empirical model of transition kernel and reward function, denoted by $\hat{P}_n(s'|s, a) = \frac{T_n(s,a,s')}{J(T)}$ and $\hat{r}_n(s, a) = \frac{1}{J(T)} \sum_{h=1}^{J(T)} r_n(s, a; h) \mathbb{1}(s_n(h) = s, a_n(h) = a), \forall (s, a, s'), n \in \mathcal{N}$, respectively. Using these empirical estimates, GM-R2MAB creates a set of plausible MDPs such that the transition

kernels and reward functions are close to the true ones, which are defined as

$$\mathcal{M} = \{M_n = (\mathcal{S}, \mathcal{A}, \tilde{P}_n, \tilde{r}_n) : |\tilde{P}_n(s'|s,a) - \hat{P}_n(s'|s,a)| \le \delta, \tilde{r}_n(s,a) = \hat{r}_n(s,a) + \delta, \forall n, s, a\}, \quad (7)$$

where $\delta = \sqrt{\frac{1}{2J(T)} \log \frac{2SANJ(T)}{\eta}}$ for $\eta \in (0,1)$ is built using the Hoeffding inequality [49].

---

**Algorithm 1** GM-R2MAB

---

**Require:** Time horizon $T$, learning function $J(T) < T$.
1: **for** $n = 1, 2, ..., N$ **do**
2:     Observe arm $n$ until there are $J(T)$ visits of pairs $(s_n(t) = s, a_n(t) = a), \forall s \in \mathcal{S}, a \in \mathcal{A}$.
3: **end for**
4: Construct the set of plausible MDPs $\mathcal{M}$ as in (7);
5: Compute the corresponding ERC index policy $\pi^\star$ by solving **ELP**($\mathcal{M}$);
6: Execute $\pi^\star$ to the end.

---

In the exploitation phase, GM-R2MAB computes the ERC index policy $\pi^\star$ by performing optimistic planning. In other words, GM-R2MAB selects an optimistic transition kernel, an optimistic reward function, and an optimistic policy to maximize the objective function while satisfying the constraints. More specifically, it can be expressed as the following optimization problem,

$$(\tilde{P} = \{\tilde{P}_n, \forall n\}, \pi^\star) = \arg\max_{M_n \in \mathcal{M}} \text{LP}(\tilde{P}_n, \tilde{r}_n, \forall n). \quad (8)$$

To solve (8), GM-R2MAB uses **Extended LP (ELP)** by leveraging the state-action-state occupancy measure $z_n(s, a, s')$ defined as $z_n(s, a, s') = P_n(s'|s,a)\omega_n(s,a)$ to express the confidence intervals of transition probabilities. The description of **ELP** is provided in Appendix A. Once we compute $\{z_n^\star, \forall n\}$, the probabilities $\xi_n(s,a)$ and hence the indices $\mathcal{I}_n(s)$ in (6) are recovered from the computed occupancy measures as $\mathcal{I}_n(s) := \frac{\sum_{s' \in \mathcal{S}} z_n^\star(s, a, s') \tilde{r}_n(s,a)}{\sum_{b \in \mathcal{A}, s' \in \mathcal{S}} z_n^\star(s, b, s')}$, from which we can construct our ERC index policy $\pi^\star$, and execute $\pi^\star$ to the end.

#### 4.1.1 Regret Analysis of GM-R2MAB

To characterize the regret, we first introduce the definition of ergodicity coefficient [4, 2]. For ease of readability, we denote the state for all arms as a stacked vector $\mathbf{s} \in \mathcal{S}^N := [s_1, s_2, \ldots, s_N]$, the corresponding actions under policy $\pi^\star$ as $\pi^\star(\mathbf{s})$, and the unknown MDPs as $\Theta := [\theta_1, \theta_2, \ldots, \theta_N]$ with $\theta_n := (P_n, r_n)$. The transition kernel of the stacked system is then $P_\Theta(\cdot|\mathbf{s}, \pi^\star(\mathbf{s})), \forall \mathbf{s} \in \mathcal{S}^N$.

**Definition 3.** $D_{P_\Theta} := 1 - \min_{\mathbf{s},\mathbf{s}'} \sum_{\mathbf{z} \in \mathcal{S}^N} \min\{P_\Theta(\mathbf{z}|\mathbf{s}, \pi^\star(\mathbf{s})), P_\Theta(\mathbf{z}|\mathbf{s}', \pi^\star(\mathbf{s}'))\}$ *is defined as the ergodicity coefficient of $P_\Theta$, and $D := \sup_\Theta D_{P_\Theta}$ as the maximum value.*

**Theorem 2.** *The regret of* GM-R2MAB *with $J(T) = \mathcal{O}(T^{1/2})$ satisfies:*

$$Reg(\pi^\star, T) = \mathcal{O}\left(\sqrt{T}\left(SAB + \frac{BN}{1-D}\sqrt{\log\frac{4SANT}{\eta}}\right)\right). \quad (9)$$

The regret comes from the exploration and exploitation phases in GM-R2MAB, respectively. Specifically, the first term $\mathcal{O}(SAB\sqrt{T})$ in (9) is the worst regret from explorations of each state-action pair under the generative model with $J(T)$ time steps for sampling, and the second term comes from the policy execution phase of GM-R2MAB due to MDP model mismatch. The proof of Theorem 2 differs from the traditional analysis framework of generative based RL for restless bandits [62, 65], particularly in the way to track regrets due to model mismatch. We leverage the form of Bellman equations of long-term average MDPs and transfer the regret to the difference of relative value functions. This enables us to track the regret relying only on the first moment behavior of transition kernels. However, [62] depended on higher order moment behavior of transition kernels to track regrets. The higher moment behavior is typically hard to analyze for a general MDP other than the birth-and-death process considered in [62]. [65] tracked regrets by leveraging the optimistic properties of a linear system, which is only applicable to the finite-horizon setting considered in [65].

**Remark 5.** We emphasize that although GM-R2MAB has a similar form as the "explore-then-commit" policy [26], a key differentiator between our GM-R2MAB and state-of-the-art methods stems from two perspectives. First, GM-R2MAB has access to a generative model and samples are taken initially to

---
**Algorithm 2** `UC-R2MAB`
---
**Require:** Initialize $C_n^0(s,a) = 0$, and $\hat{P}_n^0(s'|s,a) = 1/S$, $\forall n \in \mathcal{N}, s, s' \in \mathcal{S}, a \in \mathcal{A}$.
 1: **for** $k = 1, 2, \cdots, K$ **do**
 2:      Construct the set of plausible MDPs $\mathcal{M}^k$ as in (10);
 3:      Compute the corresponding ERC index policy $\pi^{\star,k}$ by solving **ELP**$(\mathcal{M}^k)$;
 4:      Execute $\pi^{\star,k}$ in the current episode.
 5: **end for**
---

estimate a model. As a result, `GM-R2MAB` only needs to solve an ELP once to construct a policy, which is computationally efficient. This is in contrast to the state-of-the-art colored-UCRL2 [53], which needs to solve a recursive Bellman equation to derive the policy. One contribution here is to determine the right choice of $J(T)$, which plays a key role in balancing the tradeoff between model accuracy and complexity. Second, `GM-R2MAB` executes our proposed provably optimal ERC index policy in the exploitation phase rather than using a heuristic one as in state of the arts [45, 60, 46, 59]. This contributes to the polynomial prefactor in the regret, compared to an exponential one in colored-UCRL2. Finally, we note that such a generative model based approach has also been used in the context of CMDPs [29, 30] and restless bandits [62, 65]. However, they considered either a finite-time [29, 65] or a discounted setting [30], and hence cannot be directly applied to the infinite-horizon average-reward setting studied in this paper. Furthermore, [29, 30] focused on the sample complexity analysis, which is not directly translatable to the regret [22]. Though Restless-UCB [62] operated in a similar manner as our `GM-R2MAB`, it depends on the performance of an offline "black-boxed" oracle approximator in the exploitation phase, while our `GM-R2MAB` leverages an explicit and provably optimal ERC index policy.

### 4.2 `UC-R2MAB`: Online Index-Aware Reinforcement Learning

The `GM-R2MAB` approach operates in an "offline" manner in the sense that it first estimates a model by sampling every state-action pair in the system for a certain number of times, and then computes a policy to be executed throughout the exploitation phase. Unfortunately, such an "offline" approach may not be feasible in many real-world applications since some states may not be reachable without executing the policy. To this end, we further develop an "online" approach, dubbed as `UC-R2MAB`, via interleaving the process of collecting samples from the environments and model updates, which is summarized in Algorithm 2. Specifically, `UC-R2MAB` operates in an episodic manner where each episode consists of $H$ consecutive frames. Let $K$ be the total number of episodes until time $T$, hence we have $T = KH$. Denote the $k$-th episode as $\mathcal{H}_k$ and let $\tau_k$ be the time when it starts.

At the beginning of each episode, i.e., $\tau_k, \forall k$, `UC-R2MAB` estimates the true transition kernel and the true reward by the corresponding empirical averages as $\hat{P}_n^k(s'|s,a) = \frac{C_n^{k-1}(s,a,s')}{\max\{C_n^{k-1}(s,a),1\}}$, $\hat{r}_n^k(s,a) = \frac{1}{\max\{C_n^{k-1}(s,a),1\}} \sum_{\tau=1}^{k-1} \sum_{h=1}^{H} r_n(s,a) \mathbb{1}(s_n^\tau(h) = s, a_n^\tau(h) = a)$, where $C_n^{k-1}(s,a)$ is the number of visits to state-action pairs $(s,a)$ until $\tau_k$, and $C_n^{k-1}(s,a,s')$ is the number of transitions from $s$ to $s'$ under action $a$, satisfying $C_n^k(s,a) = C_n^{k-1}(s,a) + \sum_{h=1}^{H} \mathbb{1}(s_n^k(h) = s, a_n^k(h) = a)$, and $C_n^k(s,a,s') = C_n^{k-1}(s,a,s') + \sum_{h=1}^{H} \mathbb{1}(s_n^k(h+1) = s'|s_n^k(h) = s, a_n^k(h) = a)$, $\forall(s,a) \in \mathcal{S} \times \mathcal{A}$ and $\forall(s,a,s') \in \mathcal{S} \times \mathcal{A} \times \mathcal{S}$, $\forall n$, where $s_n^k(h)$ is the state of arm $n$ at the $h$-th time frame in episode $k$.

Similar to `GM-R2MAB`, `UC-R2MAB` creates a set of plausible MDPs using these empirical estimates for each episode as in (7) by replacing $J(T)$ with $C_n^{k-1}(s,a)$ for each arm $n$. Thus for $\eta \in (0,1)$, the set of plausible MDPs in the $k$-th episode is defined as

$$\mathcal{M}^k = \{M_n^k = (\mathcal{S}, \mathcal{A}, \tilde{P}_n^k, \tilde{r}_n^k) : |\tilde{P}_n^k(s'|s,a) - \hat{P}_n^k(s'|s,a)| \le \delta_n^k(s,a), \tilde{r}_n^k(s,a) = \hat{r}_n^k(s,a) + \delta_n^k(s,a)\}. \quad (10)$$

where $\delta_n^k(s,a) = \sqrt{\frac{1}{2C_n^{k-1}(s,a)} \log \frac{4SAN(k-1)H}{\eta}}$ is built using the Hoeffding inequality [49].

Once constructing the model, `UC-R2MAB` computes the ERC index policy $\pi^{\star,k}$ in episode $k$ by solving an ELP, which is described in Appendix A. Specifically, it solves the following optimization problem,

$$(\tilde{P}^k = \{\tilde{P}_n^k, \forall n\}, \pi^{\star,k}) = \arg\max_{M_n^k \in \mathcal{M}^k} \text{LP}(\tilde{P}_n^k, \tilde{r}_n^k, \forall n), \quad (11)$$

from which it recovers the indices $\mathcal{I}_n^k(s)$ to construct ERC and then execute the index policy to the end of this episode. This is the same problem as in (8) except for substituting $\mathcal{M}$ with $\mathcal{M}^k$.

This algorithm draws inspiration from the infinite-horizon algorithm UCRL [36], which uses the sampled trajectory of each episode to update the plausible MDPs of next episode. The major difference that distinguishes our UC-R2MAB is that UC-R2MAB deploys the proposed ERC at each episode, and thus results in solving a low-complexity ELP, which is exponentially better than that of UCRL-based algorithm [36] that needs to solve extended value iterations.

#### 4.2.1 Regret Analysis of UC-R2MAB

**Theorem 3.** *The regret of* UC-R2MAB *satisfies:*

$$Reg(\{\pi^{\star,k}, \forall k\}, T) = \mathcal{O}\left(\sqrt{T}\left(\frac{1}{2} + \frac{2\sqrt{2}B\sqrt{N}}{1-D}\sqrt{\log\frac{4SANT}{\eta}}\right)\right). \tag{12}$$

Though the design of UC-R2MAB is inspired by UCRL type algorithms, the regret analysis of UC-R2MAB differs from colored-UCRL2 [53], a state-of-the-art method for online restless bandits. The major difference comes from the fact that UC-R2MAB leverages the relative value function of Bellman equation for long-term average MDPs to track regrets caused by model mismatch as GM-R2MAB. There are also recent Thompson sampling based results [2] on characterizing Bayesian regret for RMAB while using techniques reminiscent of UC-R2MAB. The sub-linear regret in [2] was achieved by upper bounding the number of episodes to be $\sqrt{T}$ under a specific requirement for the length of each episode. In contrast, UC-R2MAB directly bounds the sum of regret from each episode without such a episode length constraint.

**Remark 6.** Similar to the state-of-the-art colored-UCRL2 [53] and Thompson sampling based algorithms [37, 38, 2], our UC-R2MAB also operates in an episodic manner and achieves an $\tilde{\mathcal{O}}(\sqrt{T})$ regret. However, the colored-UCRL2 is known to be computationally expensive and the regret is exponential in the number of arms and states. Likewise, Thompson sampling based methods provide theoretical guarantees in the Bayesian settings and need to implement a computationally expensive method to update the posterior beliefs due to the complex likelihood functions. In contrast, our UC-R2MAB is computationally appealing since it only needs to solve an ELP in each episode, from which UC-R2MAB constructs and implements our proposed ERC index policy. The regret of UC-R2MAB is polynomial in the number of arms and states, which is exponentially better than that of colored-UCRL2. Finally, we remark that the multiplicative "pre-factor" that goes with the time dependent function in the regret of UC-R2MAB is smaller than that of GM-R2MAB. This is due to the fact that UC-R2MAB operates in an episodic manner while GM-R2MAB only samples once initially. This leads to a higher computation complexity of UC-R2MAB, which is $\mathcal{O}(NSAK)$ compared to that of GM-R2MAB, which is $\mathcal{O}(NSA)$.

## 5 Experiments

In this section, we present some of our experimental results. We also demonstrate the utility of our GM-R2MAB and UC-R2MAB by evaluating them under two real-world applications of restless bandits.

### 5.1 Experiments on Constructed Instance

**Instance construction.** We consider a setup with 10 classes of arms. The state space is $\mathcal{S} \in \{0, 1, 2, 3, 4, 5, 6, 7, 8, 9, 10\}$. Class-$n$ arm arrives with rate $3n$ for $n = 1, \cdots, 10$, and departs with a fixed rate of $\mu = 20$. Since Whittle index policy is only designed for RMAB with two actions, we evaluate our algorithms under two settings: (1) two-action setting; and (2) multi-action setting. For two-action setting, we consider a controlled Markov process in which states evolve as a specific birth-death process, i.e., state $s$ only transit to $s+1$ or $s-1$ with probability $P(s, s+1) = \lambda/(\lambda + \mu)$ and $P(s, s-1) = \mu/(\lambda + \mu)$. For active action, class-$n$ arm generates a random reward $r_n(s) \sim Ber(sp_n)$, with $p_n$ uniformly sampled from $[0.01, 0.1]$. The activation budget is set to $0.3N$, where we vary the number of arms $N$ from 50 to 350. For the multi-action setting, we consider a total of $3, 5, 10$ actions, and the state of arm $n$ transition to any state in $\mathcal{S}$ with a randomized non-zero transition probability. We only present results with 10 actions and provide other results in Appendix E.

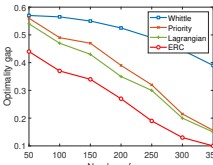 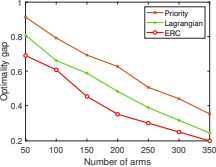 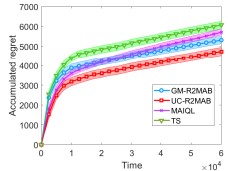 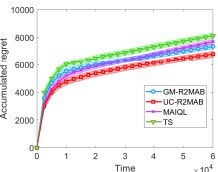

Figure 1: Optimality gap for two actions.

Figure 2: Optimality gap for 10 actions.

Figure 3: Regret for two actions.

Figure 4: Regret for 10 actions.

**Baselines.** We compare our ERC index policy with Whittle index policy (Whittle) [64], a priority policy (Priority) [61] and a lagrangian based policy (Lagrangian) [40]. We compare our RL solutions GM-R2MAB and UC-R2MAB with a Q-learning based policy (MAIQL) [39], a Thompson sampling policy (TS) [38], Restless-UCB [62], and a set of learning Whittle index based policy including Fu [25], AB [8] and NeurWin [50]. Note that we do not include results of colored-UCRL2 in our comparions since it is known to suffer from a high computational complexity and is outperformed by existing policies, e.g., Restless-UCB [62]. For ease of readability, we only present results of two "online" baselines for comparisons. More experimental results and the parameter settings for baselines are provided in Appendix E.

**Asymptotic optimality.** We compare the rewards obtained by an index policy with that from the theoretical upper bound obtained by solving the LP (2)-(5). We call the ratio between this award difference and the number of arms as the *optimality gap*. From Figures 1 and 2, we observe that all policies are asymptotically optimal, which is consistent with their theoretical performance guarantees. Our ERC slightly outperforms these baselines in terms of the vanishing speed of optimality gap.

**Regret and running time.** The accumulated regrets are presented in Figures 3 and 4, where we use the Monte Carlo simulation with $1,000$ independent trials of a single-threaded program on AMD Ryzen 5800x desktop with 64GB RAM. For simplicity, we choose 200 arms and a time horizon of $T = 60,000$ slots. Each episode consists of $2,500$ slots. We observe that UC-R2MAB achieves the lowest cumulative regret and is better than our "offline" method GM-R2MAB. This is consistent with our theoretical analysis (see Remark 6). A key observation is that for a large horizon $T$, our "offline" GM-R2MAB even outperforms the "online" MAIQL. This is because GM-R2MAB leverages our proposed provably optimal ERC index policy in the exploitation phase. We also compare the average running time of these algorithms. For two-action (10-action) setting, the average running time of GM-R2MAB, UC-R2MAB, MAIQL and TS is 86s (144s), 308s (607s), 348s (702s) and 359s (681s), respectively. It is clear that our GM-R2MAB and UC-R2MAB are more efficient in running time. These improvements come from the intrinsic design of our algorithms that merely need to solve an ELP, while TS needs to solve Bellman equations.

### 5.2 Experiments on Real-World Datasets

We demonstrate the utility of GM-R2MAB and UC-R2MAB by evaluating them under two recently studied applications of restless bandits: wireless scheduling with two actions, and tuberculosis care with multiple actions. Due to space constraints, we relegate the detailed descriptions of these two problems to Appendix E.

**Wireless scheduling over fading channels** [1]. A wireless client is modeled as an arm, which has some data to transmit. Each arm suffers from 1 unit holding cost in each time slot until the data is transmitted. The quality of wireless channel, either good or bad, via which data is transmitted determines the amount of transmitted data and varies over time. The goal is to maximize the negative of total holding cost. We adopt the settings in [1], where 1 out of 10 arms is activated at any moment in time. The regret is shown in Figure 5. It is observed that UC-R2MAB outperforms all baselines. Importantly, UC-R2MAB has a sub-linear regret guarantee while the best-performing baseline MAIQL lacks of finite-time analysis. Since the environment is dynamically changing over time, it may be hard to build a perfect simulator and hence UC-R2MAB is preferable than the offline GM-R2MAB.

**Tuberculosis care in India** [40]. A health worker takes three actions on 200 patients to improve their adherence over a course of 6 months. Each action has varying cost and effectiveness: cheap (call patients), semi-expensive (visit patients) and very expensive (escalate patients). The goal is to maximize patients' adherence subject to a daily time budget due to the limited worker time

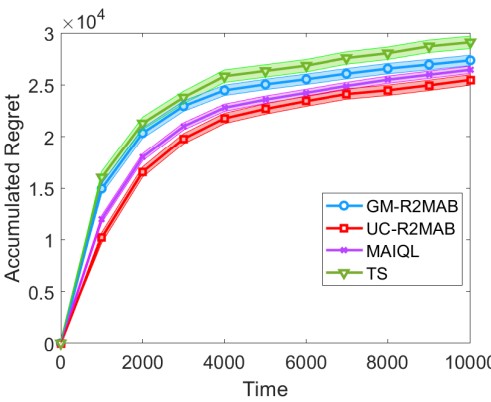

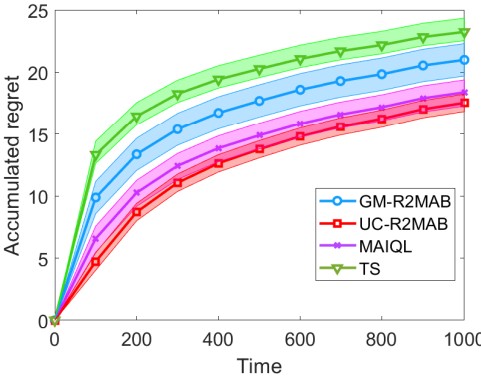

Figure 5: Wireless scheduling.         Figure 6: Tuberculosis care in India.

and resources. We adopt the settings in [40] with the budget being 20 and the reward defined as adherence level/3. From Figure 6, we again observe that `UC-R2MAB` achieves a sub-linear regret and outperforms all baselines.

## 6 Conclusion

In this paper, we developed two low-complexity index-aware RL algorithms, `GM-R2MAB` and `UC-R2MAB` for online infinite-horizon average-reward restless multi-action bandits. We proved that both algorithms achieved a sub-linear regret that is only polynomial in the number of arms and states. Our key design to reduce both the computational complexity and exponential factor in regret analysis is via exploiting the inherent structure encoded in restless bandits and leveraging our proposed provably optimal index policies for decision-makings.

## Acknowledgements

This work was supported in part by the National Science Foundation (NSF) grants CRII-CNS-NeTS-2104880 and RINGS-2148309, and was supported in part by funds from OUSD R&E, NIST, and industry partners as specified in the Resilient & Intelligent NextG Systems (RINGS) program, as well as the U.S. Department of Energy (DOE) grant DE-EE0009341. Any opinions, findings, and conclusions or recommendations expressed in this material are those of the authors and do not necessarily reflect the views of the funding agencies.

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

## Ethics Statement and Societal Impacts

Our research shows how our proposed two low-complexity index-aware RL algorithms, `GM-R2MAB` and `UC-R2MAB` perform in the setting of online infinite-horizon average-reward restless multi-action multi-armed bandits. Our main contributions are primarily analytic in nature, i.e., mainly in the theory part. The evaluation of our algorithms are conducted through a combination of mathematical analysis (e.g., finite-time analysis) and simulations. For sake of exposition and reproducibility, we leveraged a public dataset of the TB care in India [40], which are interpreted and leveraged without specialist medical-care domain knowledge, and without private human information (patients are divided into four types with a ratio, and other parameters are synthetic). However, the proposed methods are potentially relevant to any scientific application that can be formulated as a `R2MAB` framework. As for societal impact of our work, we highlight the need for specific information about involved individuals, or network metadata, which may lead to privacy issues and we hope to raise awareness of these potential issues of privacy.

One limitation of the method may come from the above discussions regarding the technical assumption of "global attractor" to prove the asymptotic optimality of `ERC` index policy. Though this is a standard assumption and widely used in the literature, it is hard to be established analytically. A possible direction or an open problem is to establish a sufficient condition to rigorously establish the global attractor property.

