# OpenReview forum: "Learning Infinite-Horizon Average-Reward Restless Multi-Action Bandits via Index Awareness"
_NeurIPS.cc/2022/Conference — NeurIPS 2022 Accept_

### Official Review · Reviewer_2ySi · 2022-06-23

**Rating:** 7
**Confidence:** 4
**Soundness:** 4 excellent
**Presentation:** 3 good
**Contribution:** 3 good

**Summary:**

The paper proposes two reinforcement learning algorithms for the multi-action multi-armed restless bandits' framework: GM-R2MAB (offline learning) and UC-R2MAB (online learning). An index-based policy is advocated for to avoid the dimensionality issue with increasing arms' count. A linear program (LP) is described to find a value upper bound for the R2MAB, and to characterize the asymptotic optimality of the ERC index-based approach. When the transition kernel and reward function are unknown, regret bounds were proven for both RL algorithms, and a comparison against other RMAB learning algorithms is provided on two experiment cases. The online learning algorithm (UC-R2MAB) was shown to give the lowest empirical regret in all cases.

**Questions:**


1. Both GM-R2MAB and UC-R2MAB are able to learn index policies without the indexability condition (needed for the Whittle index). However, if the considered RMAB is indexable, how would that affect the two algorithms' regret bound? would indexability give a tighter regret bound?

2. Assuming the RMAB is indexable, is there a characterization of the two algorithms' learned indices and the Whittle index if it exists? How would the learned index relate to the Whittle index?

3. In the checklist under 4.d, I couldn't locate in the main text or the appendix where consent was obtained for the used data, which should be explicitly mentioned in the revised manuscript.

**Limitations:**

**Limitations:** While the proposed approach is direct at multi-action RMABs and not the binary action case, the paper does not discuss the two algorithms' performance when the restless arms are indexable. A brief discussion of how indexability affects regret upper bounds would be interesting.

**Societal impact:** The paper presents two algorithms for RMABs in the average rewards' setting, including one medical trial case in India. Hence, an application-specific analysis should be performed by those wishing to use either algorithm (especially in medical trials' cases).

**Strengths And Weaknesses:**

### Strengths:

The paper contributes well in the finite-time analysis of R2MABs in the average reward case, and offers detailed theoretical regret analysis for both the offline (GM-R2MAB) and online (UC-R2MAB) algorithms. The introduction also explains in detail the lack of understanding of R2MABs' regret performance. The experiments include results from recent RMAB learning algorithms, and the experiment cases cover both the two-actions and multiple actions' cases. The necessary regret proofs (for theorems 2 and 3) provided in the appendices form the bulk of the paper's contribution and are well articulated.


### Weaknesses:

I find lemma 1 statement to be vague and the proof a bit general. The lemma statement depends on the value definition $V$ that is only mentioned in the appendix with the scaling parameter $\rho$ (for theorem 1). I find having an explicit definition of the average reward value necessary before going into the proofs. For lemma 1 proof, the wording should be modified to indicate that the original problem's feasible region $\Gamma$ is reachable for all points of the relaxed problem's region $\Gamma^/$.

In the experiments' section, I understand that the page limit prevents a longer description of the results. In the revised manuscript, the author(s) should dedicate more explanation as to why the two algorithms perform better than the baselines. Currently the discussion rightly mentions that the two algorithms solve an ELP, but it would help if the author(s) discuss UC-R2MAB and GM-R2MAB limitations compared to the baselines. In addition, it would be nice to aggregate all algorithms' results into one plot rather than splitting them between two plots (one in the main text and the other in the appendix). I currently need to look at two graphs that consider a single case study to understand how the two algorithms perform.

While the paper's technical analysis and sections are well-written, the paper's organization\spelling accuracy can be significantly improved as I highlight below:

Starting in line 19 and throughout the paper: *Restless multi-armed bandits (RMAB)...*
should be RMABs throughout the paper when the plural bandits is used.

Line 29 *"This is restrictive since the decision maker in many applications often has access to multiple actions for each arm."*
The author(s) should provide examples or cite work for this claim in the introduction.

Line 31 *"which we call the restless multi-action bandits, dubbed as R2MAB"*
the full name would be better here: multi-action multi-armed bandits.

Line 46 *Second, existing RL algorithms with theoretical guarantee of  a sub-linear...*
algorithms with **a** theoretical guarantee.

Line 58 *[57] achieved low-complexity... and not easy to be directly generalized*
**a** low-complexity policy... and **is** not easy.

Line 123 *In the remaining of the paper,*
In the remainder of the paper.

Both remarks 3 (line 237) and 4 (line 298) can be written as normal non-italicized paragraphs for easier reading.

Line 288 at the end of equation 12 (theorem 3), it should be a period and not a comma.

Line 301 *and the regret is exponentially in the number of arms and states*
and the regret is **exponential** in the number...

Line 320 *as a specific birth-and-death process*
better to call it **birth-death** process for consistency.

Line 324 *For multi-action setting*
For **the** multi-action setting

Line 337 *between this award difference and the number of arms as optimality gap.*
between this *reward* difference and  the number of arms as **the** optimality gap.

Line 351 *It is clear that our GM-R2MAB and UC-R2MAB is more efficient in running time.*
It is clear that our ... **are** more efficient...

Line 372 *Each action has varying cost and effective*
should be *effectiveness*.

As a minor note, the author(s) should also cite the paper in the introduction section:
Francisco Robledo, Vivek Borkar, Urtzi Ayesta, and Konstantin Avrachenkov. 2022. QWI: Q-learning with Whittle Index. SIGMETRICS Perform. Eval. Rev. 49, 2 (September 2021), 47–50. https://doi.org/10.1145/3512798.3512816



---------------------------------------------
### After rebuttal:

I thank the author(s) for responding to all of my comments. I read their responses and I find it to answer my concerns. In the updated paper, the author(s) should include the revised description of lemma 1 and the TB care dataset usage.

I have updated the soundness score from 3 to 4 to reflect the updates the author(s) provided. I will keep my overall score at 7.

---

> ### Author Response · Authors · 2022-08-02
> **Author Response to Reviewer 2ySi**
>
> Thank you very much for your review and constructive comments, as well as giving the positive rating of our work. Here we would like to address the reviewer's concerns and hope that can help raise the rating of our paper. The detailed responses are as follows:
>
> **Your comment:** “I find lemma 1 statement to be vague and the proof a bit general. The lemma statement depends on the value definition  V  that is only mentioned in the appendix with the scaling parameter  ρ  (for theorem 1). I find having an explicit definition of the average reward value necessary before going into the proofs. For lemma 1 proof, the wording should be modified to indicate that the original problem's feasible region Γ  is reachable for all points of the relaxed problem's region Γ/.”
>
> **Our response:**  Thank you for this insightful comment and pointing out this clarity issue.  The logic of Lemma 1 goes as follows. We have the original R2MAB in (1) (between lines 120-121 in the main paper).   Then, we relax the “hard” constraint in the original problem to be an averaged constraint, and formulated a relaxed problem as in A.1 in the supplementary material (between lines 8-9 in the supplementary material). Therefore, the optimal value achieved by this relaxed problem is an upper bound of that of the R2MAB (1) since the constraint of this relaxed problem expands the feasible region of the original R2MAB (1), i.e., the original problem (1)’s feasible region is a subset of the above relaxed problem’s feasible region due to the relaxation.
>
> In addition, as shown in [3], the above relaxed problem can be equivalently reformulated as the LP in (2)-(5) using the definition of occupancy measures \{\omega_n(s,a)\}.  Due to the equivalence between the LP (2)-(5) and the above relaxed problem, and the fact that the optimal value achieved by the relaxed problem is an upper bound of that of the R2MAB (1), we reach the conclusion in Lemma 1 that the optimal value achieved by the LP (2)-(5) is an upper bound of that of the R2MAB (1).  We are sorry that this is not clear, partly due to the space constraints.  We can move the discussions in A.1 in the supplementary material to the main paper since we will have one additional content page for the camera-ready version.

---

> > ### Author Response · Authors · 2022-08-02
> > **Author Response to Reviewer 2ySi**
> >
> > **Your comment:** “In the experiments' section, I understand that the page limit prevents a longer description of the results. In the revised manuscript, the author(s) should dedicate more explanation as to why the two algorithms perform better than the baselines. Currently the discussion rightly mentions that the two algorithms solve an ELP, but it would help if the author(s) discuss UC-R2MAB and GM-R2MAB limitations compared to the baselines. In addition, it would be nice to aggregate all algorithms' results into one plot rather than splitting them between two plots (one in the main text and the other in the appendix). I currently need to look at two graphs that consider a single case study to understand how the two algorithms perform.”
> >
> > **Our response:** Thank you for your insightful comments and suggestions.  First, as we discussed in the introduction (lines 44-47), though Whittle index policy is a celebrated heuristic for RMAB, finding it is typically intractable since the Whittle index policy is “well-defined” or “feasible” only if a so-called “indexability” condition is satisfied, which is hard to establish.  In contrast, we circumvent this limitation by developing a more general linear programming approach, and hence our ERC index policy is well defined with a low complexity.  In addition, we focus on a multi-action setting, i.e., R2MAB, while the Whittle index policy is defined  for conventional RMAB with two actions.
> >
> > Second, we mainly discussed the limitations of existing learning based algorithms for RMAB in the introduction section (lines 39-59).  The main contribution of this paper is then to develop low-complexity learning algorithms for R2MAB (a generalization of RMAB) with order-of-optimal regret.  For example, our UC-R2MAB only needs to solve LP, compared to state-of-the-art colored-UCRL2, which is much more computationally efficient.  These are the merits of our proposed algorithms compared to these baselines.  To the best of our knowledge, some baselines lack finite-time performance analysis.
> >
> > We fully understand the reviewer’s concern and we are sorry that we have to relegate some experimental results to the supplementary material due to space constraints. Here we would like to make a clarification:  As we study the general restless multi-action multi-armed bandits problem (R2MAB), we consider the performance impact of the number of actions.  This is different from the classical RMAB which only has 2 actions.  To this end, we consider 2, 3, 5, and 10 actions.  For each performance metrics, we consider these 4 cases.
> >
> > For example, Figure 1 (2 action) and Figure 2 (10 actions), as well as Figure 1 (3 actions) and Figure 2 (5 actions) in the supplementary materials are all for the evaluation of “asymptotic optimality”, which all deliver the same message that the index policies are asymptotically optimal.  We decided to choose two cases in the main paper: 2 actions, since this is the conventional setting for RMAB, and 10 actions, since this is one example for multiple actions setting. Similar observations or conclusions can be made for cases with 3 and 5 actions and hence are relegated to the supplementary materials. These plots are parallel to each other  and hence cannot be aggregated into one plot.  Similar reasons hold for Figure 3 (2 action) and Figure 4 (10 actions), as well as Figure 5 (3 actions) and Figure 6 (5 actions) in the supplementary materials for the regret comparison.
> >
> > Lastly, as mentioned above, in the classical RMAB, there are just 2 actions, and there are several baselines designed to learn Whittle index policy (2 actions), and hence we compare all in Figures 7 and 8 in the supplementary material (see Lines 244-263).  Since these two plots are in the same setting, they can be aggregated into one plot.  Per the reviewer’s approval, we can move some of these results to the main paper since we will have an additional content page for the camera-ready version.
> >
> > **Your comment:** “While the paper's technical analysis and sections are well-written, the paper's organization\spelling accuracy can be significantly improved as I highlight below:”
> >
> > **Our response:**  We thank the reviewer very much for your patience to read the paper and pointing out these typos.  We have addressed these typos in the paper and we will carefully go through the paper to further improve its quality in the camera-ready version.

---

> > > ### Author Response · Authors · 2022-08-02
> > > **Author Response to Reviewer 2ySi**
> > >
> > > **Your comment:** “Both GM-R2MAB and UC-R2MAB are able to learn index policies without the indexability condition (needed for the Whittle index). However, if the considered RMAB is indexable, how would that affect the two algorithms' regret bound? would indexability give a tighter regret bound?”
> > >
> > > **Our response:** Thank you for your insightful comments and pointing out this clarity issue.  The indexability or non-indexability is a property that is  needed for designing index policies for conventional approaches.  For example, if a problem is non-indexable, then the Whittle index policy is not feasible.  Also, the indexability is defined by Whittle in the seminal paper [61] for the classical RMAB (i.e., two actions).  To the best of our knowledge, there is no rigorous definition of indexability in the multi-action settings.   To circumvent this limitation/difficulty, we propose a more general linear programming approach to design index policies without the indexability requirement (also see Remark 1, Lines 168-170). In other words, if we consider a 2-action setting, where R2MAB reduces to RMAB, then our ERC index policy is always feasible no matter whether the underlying RMAB is indexable or non-indexable.  However, the Whittle index policy is only feasible if the RMAB is indexable.
> > >
> > > We designed GM-R2MAB and UC-R2MAB for online R2MAB, which is quite challenging, and we listed three challenges/limitations of state-of-the-art methods in the introduction (Lines 41-61).  Our key contribution is that we advocate index-aware reinforcement learning (RL) solutions to design RL algorithms operating on a much smaller dimensional subspace by exploiting the inherent structure in restless bandits.  To achieve this, we first need to design ERC index policy to exploit the inherent structure in R2MAB to address the dimensional concerns, and then two learning algorithms on top of our ERC index policy, i.e., GM-R2MAB and UC-R2MAB leverage ERC index policy to make decisions, rather than contending directly with the extremely high-dimensional state-space for decision making (lines 48-53), e.g., via repeatedly solving complicated Bellamn equations as in existing approaches for making decisions.  To this end, the indexability does not affect our learning algorithms since our index policy is well-defined regardless of indexablity.
> > >
> > > **Your comment:** “Assuming the RMAB is indexable, is there a characterization of the two algorithms' learned indices and the Whittle index if it exists? How would the learned index relate to the Whittle index?”
> > >
> > > **Our Response:** As we are considering general RMAB with multiple actions, there is hardly an argument related with indexability. Indexability condition is established upon the binary action setting, where a=0 represents passive action and a=1 as active action, and thus Whittle’s index is defined for binary-action RMAB. To the authors’ best knowledge, indexability conditions related to multiple actions have not been well defined yet. In this sense, it is difficult to claim any connection between whittle’s index and our proposed indices for the general RMAB setting.
> > >
> > > Nevetless, when considering RMAB with binary action setting and indexability condition is satisfied, the Whittle index is also asymptotically optimal and thus it belongs to the category of the proposed index policies. For example, we numerically verify the asymptotic optimality of these index policies in Figure 1 with 2 actions, i.e., the classical RMAB setting.
> > >
> > > **Your comment:** “In the checklist under 4.d, I couldn't locate in the main text or the appendix where consent was obtained for the used data, which should be explicitly mentioned in the revised manuscript.”
> > >
> > > **Our response:** Thank you for the comment. The data set we used is a public data set as mentioned in [37], which are interpreted and leveraged without specialist medical-care domain knowledge, and without private human information (patients are divided into four types with a ratio, and other parameters are synthetic). We follow the same setting as [37] for ease of exposition and reproducibility.

---

> > > > ### Author Response · Authors · 2022-08-02
> > > > **Author Response to Reviewer 2ySi**
> > > >
> > > > **Your comment:** “Limitations: While the proposed approach is direct at multi-action RMABs and not the binary action case, the paper does not discuss the two algorithms' performance when the restless arms are indexable. A brief discussion of how indexability affects regret upper bounds would be interesting."
> > > >
> > > > **Our response:** As we are considering general RMAB with multiple actions, there is hardly an argument related with indexability. Indexability condition is established upon the binary action setting, where a=0 represents passive action and a=1 as active action, and thus Whittle’s index is defined for binary-action RMAB. To the authors’ best knowledge, indexability conditions related to multiple actions have not been well defined yet. In this sense, it is difficult to claim any connection between whittle’s index and our proposed indices for the general RMAB setting.
> > > >
> > > > Nevertheless, when considering RMAB with binary action setting and indexability condition is satisfied, the Whittle index is also asymptotically optimal and thus it belongs to the category of the proposed index policies.
> > > >
> > > >
> > > > **Your comment:** “Societal impact: The paper presents two algorithms for RMABs in the average rewards' setting, including one medical trial case in India. Hence, an application-specific analysis should be performed by those wishing to use either algorithm (especially in medical trials' cases)."
> > > >
> > > > **Our response:** Our research shows how the proposed two low-complexity index-aware RL algorithms, GM-R2MAB and UC-R2MAB perform for online infinite-horizon average-reward restless multi-action bandits theoretically and numerically. For the sake of exposition and reproducibility, we have used public dataset of the TB care in India [37], which are interpreted and leveraged without specialist medical-care domain knowledge, and without private human information (patients are divided into four types with a ratio, and other parameters are synthetic). However, the proposed methods are potentially relevant to any scientific application that can be formulated as a R2MAB framework. As for societal impact of our work, we highlight the need for specific information about involved  individuals, or network metadata, which may lead to privacy issues and we hope to raise awareness of these potential issues of privacy.

---

### Official Review · Reviewer_4AZS · 2022-07-03

**Rating:** 7
**Confidence:** 3
**Soundness:** 4 excellent
**Presentation:** 4 excellent
**Contribution:** 3 good

**Summary:**

The authors address infinite horizon average reward restless multi-action bandits. They propose a new type of index, and two algorithms that calculate this index when the transitions and rewards are unknown - one offline and one online. The authors provide regret bounds for their algorithms and compare them empirically to other methods.

**Questions:**

The paper is very clear and I have no questions or suggestions.

**Limitations:**

Not relevant for this paper.

**Strengths And Weaknesses:**

Originality
The paper is novel to the best of my knowledge. The authors emphasize that their regret bounds are highly novel, I have no knowledge to contradict that fact.

Quality
The paper is of very high quality - the presentation of the problem is clear and well written, the regret bounds are impressive and strong,  and the experiments form a convincing argument.

Clarity
Despite the results being mostly theoretical, I found the paper to be very clear and very well written.

Significance
Restless multi-action bandits is a a bit of a niche so the significance of the paper is somewhat limited. The proposed algorithms and simulations correspond to discrete MDPs whose model is learned, which further hampers the significance of the work for applications. Inside this niche, the results do seem convincing, interesting and filling in missing knowledge.

---

> ### Author Response · Authors · 2022-08-02
> **Author Response to Reviewer 4AZS**
>
> Thank you very much for your review and constructive comments, as well as giving the positive rating of our work.

---

### Official Review · Reviewer_2MaK · 2022-07-14

**Rating:** 5
**Confidence:** 4
**Soundness:** 3 good
**Presentation:** 3 good
**Contribution:** 3 good

**Summary:**

Authors provide two learning algorithms for the multi-action restless bandit problem (R2MAB), with new regret bounds that advance state of the art. Their algorithm and bounds rely on a new index policy which is asymptotically optimal under a global attractor condition. Authors provide numerical experiments on three domains that support their claims.

**Questions:**

 - Please respond to the weaknesses above.


Note: the paper potentially has enough merit to be considered for acceptance in my view, but the points listed in "weaknesses" need to be clarified.

-------
I have read the other reviews and author responses and am satisfied with their answers to my questions, and have increased my score. Authors should add more clarifying language about the global attractor property's reliance on data distributions as is done in Verloop 2016 section 6.

**Limitations:**

 - Please see weaknesses, second comment

**Strengths And Weaknesses:**

Strengths:
 - The draft is mostly well-written, making the contributions clear.
 - Authors provide experimental results on three domains showing wins on each (though modest compared to MAIQL on scheduling and TB)
 - The authors provide attractive regret bounds or the average reward, R2MAB case, advancing current state of the art.
 - Authors introduce a new index policy class for R2MAB, ERC, which seems to have good performance and will be of general interest to the RMAB community, though it seems to rely on a numerically verifiable global attractor condition.
 - Authors provide two learning algorithms, GM-R2MAB and UC-R2MAB, where the former is "offline" in that it takes time to collect enough samples to build a confident world model, and the latter is "online" in that it follows more closely to an upper confidence bound approach, exploring and exploiting in tandem. UC-R2MAB seems to perform better than all baselines in experiments.

Weaknesses:
 - My understanding is that the global attractor property of a policy class is data dependent, and thus needs to be verified for each new dataset to which a policy class is applied. All of the authors' core results seem to depend on the global attractor condition, i.e., optimality of the ERC index policy they propose, and thus also the regret bounds (which rely on the ERC being asymptotically optimal). So, unless I am missing something, all of the results here depend on the users' ability to numerically verify the global attractor policy for the data to which they hope to apply the authors' algorithms. This may be ok, but the authors need to make it much more clear throughout the draft if this is the case. For instance, in Remark 1, authors claim that "ERC does not require the indexability condition, which is often hard to establish especially when the transition kernel of the underlying MDP is convoluted" -- while true, one seems to have to verify a global attractor condition instead to use the authors' policy, which is ultimately very similar since the indexability condition is a necessary condition for the global attractor property of the Whittle index policy, and itself is often verified numerically for new MDP classes. So authors should clearly state the tradeoff.
 - The authors mark N/A in their checklist for both limitations and social impacts -- this is not acceptable in the current iteration of NeurIPS. Authors need to engage in some amount of discussion in the limitation of their methodology. Also, since the authors present experimental results on a tuberculosis care domain, it is reasonable to ask that they discuss how their algorithm might have adverse social impact, and so they should engage in this part of the discussion as well.
 - More evaluation would be helpful. Specifically, authors vary the number of actions, but not the number of arms, which is they key scaling parameter in RMAB. Authors should provide some experiments to show both regret and runtime scaling as N increases for at least one domain. Additionally, the authors compare against MAIQL from Killian et al. 2021, but not LPQL, which seems to be the preferred R2MAB learning algorithm from that paper. Presumably they did not compare against LPQL because it handles the discounted reward case, rather than average. However, in the appendix, authors compare against NeurWIN which handles the discounted reward case. So authors should also compare against LQPL for completeness.

---

> ### Author Response · Authors · 2022-08-02
> **Author Response to Reviewer 2MaK**
>
> Thank you very much for your review and constructive comments. Here we would like to address the reviewer's concerns and hope that can help raise the rating of our paper. The detailed responses are as follows:
>
> **Your comment:** “My understanding is that the global attractor property of a policy class is data dependent, and thus needs to be verified for each new dataset to which a policy class is applied. ..."
>
> **Our response:** Thank you for this important comment. We agree with the reviewer’s suggestions. We would like to first clarify two important concepts in restless multi-armed bandits (RMAB) literature: (1) indexability, and (2) global attractor.  Then we discuss how to improve our paper based on the reviewer’s suggestions.
>
> (1) “Indexability”:  As is known, Whittle [61] proposed the celebrated heuristics called Whittle index policy for addressing the hardness of RMAB.  However, the “feasibility” or the definition of Whittle index policy is based on the condition that a so-called “indexability” property must be satisfied.  In other words, if a problem is not indexable, then the Whittle index policy is not feasible, i.e., cannot be defined and hence cannot be applied to solve that problem (i.e., applied to the dataset).  Exacerbating this challenge is the fact that establishing the indexability of RMABs is typically intractable [48], and hence Whittle indices of many practical problems remain unknown except for a few special cases.  This is due to the fact that many practical problems are naturally not indexable [56], and hence many efforts have been focused on designing index policies without the requirement of the indexability, i.e., making index policies feasible even if the problem is nonindexable, e.g., [31,64,65,62].  However, these cannot be applied to our problem.  See Remark 1, particularly lines 165-170 for discussions.
>
> In summary, “indexability” is a condition that must be satisfied in order to define or make Whittle-like policies **feasible**.  We focus on designing index policies that are always feasible without this condition, and hence can be always defined for problems that can be formulated as RMABs (or more precisely R2MAB), no matter if this problem is indexable or not.
>
> (2) “Global attractor”:  The asymptotic optimality of index policies is often shown using a fluid limit analysis by considering the regime of a large-scale system.  For instance, the seminal work [60] established the asymptotic optimality of Whittle index policy by showing that the state distribution under Whittle index policy and the steady-state distribution under the optimal policy for the corresponding relaxed problem diminishes to zero in the asymptotic regime. To this end, [60] defined a technical condition “global attractor” to prove the asymptotic optimality of Whittle index policy, which is feasible conditioned on that the indexability condition is satisfied, as discussed above.  Following [60], most existing literature, e.g., [29,58,67,23] on proving the asymptotic optimality focuses on such a fluid limit and often makes the **technical assumption** that a fixed point of the proposed index policies satisfy the global attractor condition.
>
> In summary, “global attractor” is a technical assumption that is made to prove the performance (i.e., asymptotic optimality) of index policies.  As we stated in Remark 2 (lines 182-185), though it is hard to analytically establish that a fixed point is a global attractor, we numerically show that the fixed point of our process indeed satisfies this assumption, and hence this assumption is indeed valid. (Since this is a technical assumption, many works did not even numerically verify it).
>
> We thank the reviewer again for this important suggestion.  When we characterize the regret, we leveraged the asymptotic optimality of the ERC index policy, and the proof of asymptotic optimality is based on the global attractor property.  The regret in [2] is defined in a similar manner (with respect to Whittle index policy, which is asymptotically optimal).  Part of the reason is that finding the offline optimal policy for RMAB or R2MAB is typically intractable.  We propose two ways to make this clear in the paper since we will have one additional content page for the camera-ready version.  One is to update Definition 2 of the regret (Lines 192-197), the other is to add an additional remark to illustrate how the regret is computed, similar to [2].  Per the reviewer’s approval, we will include them in the camera-ready version.

---

> > ### Author Response · Authors · 2022-08-02
> > **Author Response to Reviewer 2MaK**
> >
> > **Your comment:** “The authors mark N/A in their checklist for both limitations and social impacts ....”
> >
> > **Our response:** Our research shows how our proposed two low-complexity index-aware RL algorithms, GM-R2MAB and UC-R2MAB perform in the setting of online infinite-horizon average-reward restless multi-action multi-armed bandits.  Our main contributions are primarily analytic in nature, i.e., mainly in the theory part.  The evaluation of our algorithms are conducted through a combination of mathematical analysis (e.g., finite-time analysis) and simulations.  For sake of exposition and reproducibility, we used a public dataset of the TB care in India [37], which are interpreted and leveraged without specialist medical-care domain knowledge, and without private human information (patients are divided into four types with a ratio, and other parameters are synthetic).  However, the proposed methods are potentially relevant to any scientific application that can be formulated as a R2MAB framework. As for societal impact of our work, we highlight the need for specific information about involved  individuals, or network metadata, which may lead to privacy issues and we hope to raise awareness of these potential issues of privacy.
> >
> > One limitation of the method may come from the above discussions regarding the technical assumption of “global attractor” to prove the asymptotic optimality of ERC index policy.  Though this is a standard assumption and widely used in the literature, it is hard to be established analytically.  A possible direction or an open problem is to establish a sufficient condition to rigorously establish the global attractor property.
> >
> > We would add these statements in the camera-ready version per the reviewer’s approval.
> >
> > **Your comment:** “More evaluation would be helpful. Specifically, authors vary the number of actions, but not the number of arms, which is they key scaling parameter in RMAB. Authors should provide some experiments to show both regret and runtime scaling as N increases for at least one domain (Fig 13-16, Fig 17-18). ...”
> >
> > **Our response:** Thank you for your insightful comments and suggestions.  We have added additional numerical results to the supplemental materials.  We place them there for the sake of explanation, and we can add or replace some results in the main paper per the reviewer’s approval since we will have one additional content page for the camera-ready version.
> >
> > Asymptotic optimality: We evaluate the asymptotic optimality of index policies in consideration with the number of arms.  Similarly, we consider four cases, 2, 3, 5, and 10 actions.  In each case, we vary the number of arms up to 20,000.  From Figures 9-12 in the supplementary material (on page 12), Again, we observe that all policies are asymptotically optimal. Though the optimality is established in the asymptotic regime (i.e., a large number of arms), we observe that the optimality gap of each policy quickly decreases and gets close to zero.
> >
> > Regret and running time: We consider 2 and 10 actions, with 200 and 2,000 arms.  The corresponding accumulated regrets are shown in Figures 13-16.  We also compare our algorithms with a state-of-the-art method named LPQL [36].  Again, we observe that UC-R2MAB achieves the lowest accumulative regret.  We also compare the average running time of these algorithms.  For two-action (10-action) setting with a total of 200 arms, the average running time of GM-R2MAB, UC-R2MAB, MAIQL, LPQL, and TS is 86s (144s), 308s (607s), 348s (702s), 314s (623s) and 359s (681s), respectively.   Similarly, when the total number of arms is 1,000, the average running time with two-action (10-action) of GM-R2MAB, UC-R2MAB, MAIQL, LPQL, and TS is 114s (188s), 443s (813s), 512s (912s), 470s (947s) and 560s (901s), respectively.  Finally, when the total number of arms is 2,000, the average running time with two-action (10-action) of GM-R2MAB, UC-R2MAB, MAIQL, LPQL, and TS is 179s (261s), 703s (1187s), 810s (1354s), 724s (1247s) and 823s (1340s), respectively.  It is clear that our GM-R2MAB, and UC-R2MAB are more efficient in running time.  As pointed out by the reviewer, LPQL (multi-action setting) and NeurWIN (conventional 2-action setting) were born for discounted rewards, rather than the infinite-horizon average-reward setting considered in this paper.  To make a relatively fair comparison, we implemented them with a discount factor of 0.99 to be close to 1.
> >
> > The regret comparison using two real settings are also presented in Figures 17 and 18. We again observe that UC-R2MAB achieves a sub-linear regret and outperforms all baselines.

---

> > > ### Author Response · Authors · 2022-08-08
> > > **Follow-up**
> > >
> > > Since the reviewer-author discussion period is ending soon, we just wanted to check in and ask if our rebuttal clarified and answered your questions. We would be very happy to engage further if there are additional questions.
> > >
> > > Also, we wanted to check if our additional clarifications regarding the merits of the paper would convince the reviewer to raise the score. Thank you!

---

### Official Review · Reviewer_mAyS · 2022-07-16

**Rating:** 6
**Confidence:** 2
**Soundness:** 3 good
**Presentation:** 3 good
**Contribution:** 3 good

**Summary:**

The paper considers the online RMAB problem, with multiple available actions and when the MDP on each arm is unknown. The paper derives an index policy for the online RMAB problem, different from the WHittle index policy and then proposes two index-aware RL algorithms. The paper shows that both algorithms achieve sublinear regret.

**Questions:**

- Line 114 says w.l.o.g r(s,0)=0. However, how is this general given that r() could potentially depend on s, i.e. r(s,0) =s?
- Lines (70-74) say that the paper takes a more general approach, hence is more efficient. However, this seems a bit counterintuitive; shouldn’t there be a trade-off? Is there something else that is being given up?


**Ethics Review Area:**

["I don’t know"]

**Limitations:**

Yes

**Strengths And Weaknesses:**

**Strengths**




- The approach seems principled and theoretically solid. The paper proves that their proposed index based approach is asymptotically optimal.
 - Empirical analysis: The experiments seem to be comprehensive; they cover a wider range of settings and test against most of the relevant baselines, including fairly recent baselines.
- The context is well set up: explaining where the sota is and what gap needs to be filled.

**Weaknesses**

- The writing/exposition is not clear at some places: For instance, the abstract says each arm evolves according to a Markov chain. However, later in section 2, each arm is said to evolve according to an MDP.

- Minor issues: grammatical issues/typos involving usage of plural/singular quantity at few places

---

> ### Author Response · Authors · 2022-08-02
> **Author Response to Reviewer mAyS**
>
> Thank you very much for your review and constructive comments, as well as giving the positive rating of our work. Here we would like to address the reviewer's concerns and hope that can help raise the rating of our paper. The detailed responses are as follows:
>
> **Your comment:** “The writing/exposition is not clear at some places: For instance, the abstract says each arm evolves according to a Markov chain. However, later in section 2, each arm is said to evolve according to an MDP.”
>
> **Our response:** Thanks for pointing out this clarity issue. An MDP is a controlled Markov chain and the state transition depends on the controlled action. To be consistent, we will change the “Markov chain” in the abstract to “an MDP”.
>
> **Your comment:** “grammatical issues/typos involving usage of plural/singular quantity at few places”
>
> **Our response:** We are sorry for these typos.  We have addressed some and will carefully go through the paper again to address all typos in the final version.
>
> **Your comment:** “Line 114 says w.l.o.g r(s,0)=0. However, how is this general given that r() could potentially depend on s, i.e. r(s,0) =s?”
>
> **Our response:**  Thank you for your insightful comments and pointing out this clarity issue. In the RMAB setting, it is widely assumed that passive arms (i.e., with action a=0) lead to no reward, i.e., r(s,0)=0.  However, this reward model does not affect the policy design and its impact on bounding the regret is minor. It only has an impact on the multiplicative “pre-factor" that goes with the time-horizon dependent function in the regret, i.e., we still achieve a sub-linear regret with a polynomial prefactor.   In our simulations, positive rewards are generated only for active arms.
>
> **Your comment:** “Lines (70-74) say that the paper takes a more general approach, hence is more efficient. However, this seems a bit counterintuitive; shouldn’t there be a trade-off? Is there something else that is being given up?”
>
> **Our response:** Thank you for this insightful comment.  We respond to this question from two perspectives: (1) The feasibility of index policies (i.e., indexability or not); and (2) The performance of index policies (i.e., asymptotic optimality).
>
> ---“Feasibility”: As we discussed in the introduction (lines 44-47), though Whittle index policy is a celebrated heuristic for RMAB, finding it is typically intractable since the Whittle index policy is “well-defined” or “feasible” if a so-called “indexability” condition is satisfied, which is hard to establish.  As a result, Whittle indices of many practical problems remain unknown except for a few special cases.  Exacerbating this challenge is the fact that the Whittle index policy is defined for the conventional RMAB, i.e., only two actions (passive or active), while we consider a general RMAB with multiple actions, i.e., R2MAB.  In this paper, we bypass this issue to design index policy via a general LP approach. Our proposed framework to design index policy does not require the indexability condition, and our index policy is well defined and feasible for both indexable and non-indexable problems, with the latter extensively existing in practice (e.g., [58], and see lines 165-170).
>
> ---“Asymptotic optimality”: The asymptotic optimality of index policies is often shown using a fluid limit analysis by considering the regime of a large-scale system.  For instance, the seminal work [60] established the asymptotic optimality of Whittle index policy by showing that the state distribution under Whittle index policy and the steady-state distribution under the optimal policy for the corresponding relaxed problem diminishes to zero in the above asymptotic regime.  [60] defined a technical condition “global attractor” to prove the asymptotic optimality of Whittle index policy, which is feasible conditioned on that the indexability condition is satisfied, as discussed above. We also focus on such a fluid limit in this paper to show the asymptotic optimality of our ERC index policy under the technical condition “global attractor”.  This technical condition is difficult to establish analytically and is only verified numerically, as in many prior works [60, 29, 58, 67, 23].  See Remark 2 (lines 182-185).

---

> > ### Author Response · Authors · 2022-08-08
> > **Follow-up**
> >
> > Since the reviewer-author discussion period is ending soon, we just wanted to check in and ask if our rebuttal clarified and answered your questions. We would be very happy to engage further if there are additional questions.
> >
> > Also, we wanted to check if our additional clarifications regarding the merits of the paper would convince the reviewer to raise the score. Thank you!

---

> > ### Comment · Reviewer_mAyS · 2022-08-08
> > **Thank you for the rebuttal**
> >
> > I have read through the authors' rebuttal as well as the other reviews and have found the responses mostly satisfactory.  I'd like to keep my score unchanged.
> >
> > However, wrt the comment on line 114 "w.l.og...", I think in general, there can be a monumental difference caused by a reward function that accrues reward from passive arms vs one that doesn't. There are also several papers that consider reward from all arms (not just active). In light of this, clarifying that the model is r(s,0)=0 (not without loss of generality) would be valuable in my view.

---

> > > ### Author Response · Authors · 2022-08-09
> > > **Author Response to Reviewer mAyS**
> > >
> > > We thank the reviewer again for clarifying this problem. In the following, we further discuss the reward function in the context of restless multi-armed bandits (RMAB).  Note that we consider RMAB, more precisely, R2MAB in this paper, rather than the classical MAB (which is stateless in general), while each arm is stateful (via a MDP) in RMAB.
> > >
> > > In general, there are two reward models considered in the RMAB literature:
> > >
> > > Model 1: All arms yield reward no matter the arms are activated or not;
> > >
> > > Model 2: Only activated arms yield rewards.
> > >
> > > Both models have been widely used and have arisen in different applications.  For example,  model 1 is widely adopted for queuing problems, e.g., References [15] [16] [26] [30] in our main paper, where all queues incur holding reward (or cost), along with others e.g., References [5] [14] [24] [36] in our main paper. Model 2 is widely adopted for cognitive radios, e.g., References [9] [19] [20] [42] in our main paper, where rewards are generated only on the state of selected channels, along with many other learning augmented RMAB literatures, e.g., References [2] [56] [57] [59] [62] in our main paper. These two models are similar without fundamental differences as discussed in [2], and they are exactly the same under the assumption that r(s,0)=0.
> > >
> > > Our design of index policy and then index-aware RL algorithms are general and hold for both models with minor differences in the performance guarantees.  First, the designed ERC index policy works on both reward models. As given in equation (6), the reward model only affects the absolute values of the ERC indices, while it has no effect on the implementation of the proposed index policy. Second, the regret bound only has a marginal difference in the multiplicative prefactor that goes with the time-dependent function in the regret bound under these two models, where the difference lies between the number of arms N and the number of maximum activated arms, which is upper bounded by B. In the current setting (i.e. Model 2), the reward generated by activated arms will not exceed B due to the fact that r\in [0,1]. This gives the bound in Lemma 4, Lemma 6, Lemma 7, Lemma 10, and Lemma 15 (see supplementary materials), where B only shows in the prefactor that goes with the time-dependent function in the regret bound. If we switch to Model 1, where passive arms also generate reward, then the reward generated will not exceed N due to the fact that r\in [0,1], which changes slightly in the prefactor of regret bound. It will not affect the sublinear regret order under the square root.  Similar arguments were discussed in Reference [2] for learning the classical Whittle index policy.
> > >
> > > We thank the reviewer's valuable comment and provide us this opportunity to clarify this issue.  We believe this clarification will further improve the quality of the paper.  We will remove the w.lo.g., and instead to discuss the aforementioned difference of these two models, as well as the generalization of our proposed policies for these two models in the camera-ready version of the paper, e..g, adding a remark.

---

### Meta-Review · Area_Chair_66Ct · 2022-08-26

**Recommendation:** Accept
**Confidence:** Certain

**Metareview:**

The paper tackles the challenging problem of online learning restless multi armed bandit (RMAB) policies. Among its contributions are the introduction of a new tractable class of RMAP policies to learn over, and tractable learning algorithms, with regret guarantees, along the lines of statistical upper confidence bounds. These could serve as useful building blocks for theoreticians and practitioners in the area alike.

The contributions of the paper are unanimously acknowledged to be positive by the reviewers, after their initial reviews were responded to in detail by the paper's author(s) leading to helpful clarifications. In view of this, I recommend acceptance of the paper.

**Award:**

No

---

### Decision · Program_Chairs · 2022-09-14

Accept